# Practitioners' viewpoints on citizen science in water management: a case study in Dutch regional water resource management

Ellen Minkman[1,2,3], Maarten van der Sanden[2] and Martine Rutten[1]

[1]Department of Water Resource Management, Delft University of Technology, Delft, 2628 CN, The Netherlands
[2]Department of Science Education and Communication, Delft University of Technology, Delft, 2628 CJ, The Netherlands
[3]Presently at Department of Public Administration and Sociology, Erasmus University Rotterdam, Burgemeester Oudlaan 50, 3062 PA Rotterdam, the Netherlands

*Correspondence to*: Martine Rutten (m.m.rutten@tudelft.nl)

**Abstract.** In recent years, governmental institutes have started to use citizen science as a form of public participation. The
Dutch water authorities are among them. They face pressure on the water governance system and a water awareness gap among the general public, and consider citizen science a possible solution. The reasons for practitioners to engage in citizen science, and in particular those of government practitioners, have seldom been studied. This article aims to pinpoint the various viewpoints of practitioners at Dutch regional water authorities on citizen science. A Q-methodological approach was used because it allows for exploration of viewpoints and statistical analysis using a small sample size. Practitioners (33) at
eight different water authorities ranked 46 statements from agree to disagree. Three viewpoints were identified with a total explained variance of 67%. Viewpoint A considers citizen science a potential solution that can serve several purposes, thereby encouraging citizen participation in data collection and analysis. Viewpoint B considers citizen science a method for additional, illustrative data. Viewpoint C views citizen science primarily as a means of education. These viewpoints show water practitioners in the Netherlands are willing to embrace citizen science at water authorities, although there is no support
for higher levels of citizen engagement.

## 1 Introduction

The OECD (Organization of Economic Cooperation and Development) named the Netherlands *"an international example"* of water resource management in their 2014 report, but warns for *"a striking awareness gap among Dutch citizens related to key water management functions, how they are performed and by whom."* (OECD, 2014, p. 21) The main causes for this
awareness gap are the absence of major water calamities in the past 60 years and the improvement of water quality over the past decades. Dutch citizens take the excellent water resource management for granted (OECD, 2014), which poses social challenges for Dutch water resource management. Citizens' behaviour counteracts efforts of water authorities; flood defences are violated by property development and civic pollution is common (OECD, 2014). Citizens and interest groups do not recognise water threats (Tielrooij, 2000), which causes a decreasing support for investments in flood defence and
water quality management (OECD, 2014; Tielrooij, 2000; UvW, 2015a). Other countries also experience a "*Lack of citizen*

*concern about water policy and low involvement of water users' associations."* (OECD, 2011, p. 60). Half of the reviewed OECD countries across the globe face such challenges including Chile, Italy, Korea and Mexico (OECD, 2011, p. 61). The governing body Dutch Water Authorities (Unie van Waterschappen, UvW) concluded that collaboration with other government layers, industry, interest groups and citizens is needed (UvW, 2015a). The UvW envisions increased public participation, with citizen science as a form of such participation (UvW, 2015b). In addition to awareness raising, citizen science could contribute to data collection and help water authorities to enhance their monitoring programs particularly with respect to the Water Framework Directive.

Definitions of citizen science can be narrow and focussed solely on data collection for academic purposes. Silvertown (2009) describes citizen science in a broader perspective, applicable to citizen science in practitioners' activities considered in this article. *"Today, most citizen scientists work with professional counterparts on projects that have been specifically designed or adapted to give amateurs a role, either for the educational benefit of the volunteers themselves or for the benefit of the project. The best examples benefit both."* (Silvertown, 2009, p. 467). To prevent confusion a distinction is made within this category of professional counterparts (Silvertown, 2009). The professional counterparts in Silvertown's definition include scientists, conservation professionals and government practitioners. We define scientists as those involved in academia. Conservation professionals are those working at nature managers or conservation organisations. Government practitioners are defined as those working at a government agency or at the local government level.

Citizen science in water resource management is upcoming, but lingering on the verge of breakthrough (Buytaert et al., 2014; Cohn, 2008; Fraternali et al., 2012). The rise of robust, cheap and low-maintenance sensors enhances opportunities for citizen science in the complex arena of water resource management (Buytaert et al., 2014). New hydrological modelling frameworks (e.g. Clark et al., 2015) and specifically uncertainty quantification (e.g. Shoaib et al., 2016) can allow for more effective design of citizen science campaigns to systematically reduce model uncertainty, as well as more effective use of citizen-collected data. These data are often considered as highly uncertain by practitioners as they indicated in the exploratory interviews held at the start of this research. Fraternali et al. (2012) give an overview of the potential of amateurs taking part in data collection, data analysis and the process of decision making in water resource management.

This potential is also recognised in the Netherlands. In November 2014 water authority Delfland organized a workshop[1] on big data and citizen science in Delft, the Netherlands. Dutch regional water managers expressed their interest in citizen science during this workshop, although they also indicated they doubts regarding this approach. Their main questions were a) what motivates citizens to participate?; b) what should be the role of citizens?; c) why should a water authority engage in citizen science?.

---

[1] Part of the symposium 'De fysieke Digitiale Delta' [the Physical Digital Delta, see also www.digitaledelta.nu/en/events.

Citizens' motivations have been studied extensively in a diverse set of citizen science projects, such as online crowdsourcing (e.g. Chandler and Kapelner, 2013; Raddick et al., 2010; Rogstadius et al., 2011), environmental monitoring (e.g. Hobbs and White, 2012; Roy et al., 2012) and meteorology (e.g. Gharesifard and Wehn, 2016). It has been acknowledged that the idea of 'the public' does not exist (e.g. Varner, 2014), since 'the public' consists of a wide variety of people with different backgrounds, interests, traits, values and beliefs. Nevertheless, existing studies of citizen science in field and online projects, despite this diversity, reveal the same dominant motivations over a wide range of projects and participants. Most mentioned reasons for citizens to engage in citizen science are: because they think it is fun; because the topic interests them; and because the topic matters to them, e.g. they want to contribute to science or nature conservation (e.g. Chandler and Kapelner, 2013; Hobbs and White, 2012; Raddick et al., 2010; Rogstadius et al., 2011; Roy et al., 2012). Citizens are motivated to continue to contribute by: (increasing) the extent of their involvement (Rotman et al., 2012; Roy et al., 2012), offering feedback concerning the work at three levels (individual contribution, group contribution and the use of data) and building a relationship based on trust between scientists and citizens (Rotman et al., 2012). The importance of trust is stressed by authors in the field of water management as well (Buytaert et al., 2014; Gharesifard and Wehn, 2016).

The role of citizens varies depending on the purpose of the citizen science project. Tulloch et al. (2013) studied the purpose of citizen science projects in bird watching, a research field with a century-long history of citizen science. The most common purpose of citizen science is *knowledge generation* (knowledge also new to the involved professionals), followed by improving *monitoring methods* and *raising awareness* among citizens. In water resource management citizen science can enhance knowledge about the water system, for example by generating knowledge on different spatial or temporal scales. It may also be used to add or test new monitoring methods to the existing monitoring network (mentioned in exploratory interviews held at the start of this study) . Citizen science can also be used to increase the water awareness that was dubbed absent in the OECD report (OECD, 2014). According to Tulloch et al. (2013), citizen science can be used to *improve management* practice as well. In water resource management such improved management can be the result of more frequent monitoring. Literature mentions *public education* as an important purpose of citizen science (e.g. Cohn, 2008), but Tulloch et al. (2013) found that public education was rarely the main purpose. Other identified, yet also more rare, purposes include *doing social research* (e.g. on human behaviour); *offer recreation* and *serendipity* (i.e. unexpected discoveries). A more recent study in the field of ecology specified the potential of citizen science for the purpose of *policy development* (Hollow, Roetman, Walter, & Daniels, 2015). In early stages of a policy development process citizen science can be used to discover alternative management actions. In case there is a range of alternatives, citizen science can be used to measure the public opinion. In later stages it can be used to persuade the public opinion towards a desired alternative or to provide a legal justification for the chosen policy. Citizen science-based data can be used for decision making in water resource management as well (Macknick and Enders, 2012). Projects can have one or multiple purposes, as the iSPEX project

demonstrates. This project served both *knowledge generation, public education* and *method improvement* (Land-Zandstraet al., 2015; Snik et al., 2014).

For the purposes of knowledge generation, improving monitoring methods, improve management and policy development.
Bonney et al. (2009) provides a useful classification of citizens' roles. They suggest there are basically three levels of citizen involvement possible: contribution, collaboration and co-creation. In a *contributory project*, citizens are mainly involved in data collection, the research question and design is the done by scientists or experts. In *collaborative projects* citizens are involved in the analysis and can be involved in the design and dissemination of results as well. In *co-created projects* citizens are involved in all steps of the research process and may even initiate the project. The vast majority of studies in the overview presented by Bonney et al. (2009) considered contributory projects. Even the occasionally co-created projects were part of multi-case studies, in which contributory projects dominated the results. Citizens' involvement in activities other than data collection may serve different purposes, for example citizen-based goal-setting could enhance adaptive management practices (Cooper, Dickinson, Phillips, & Bonney, 2007). The classification of Bonney et al. (2009) is frequently cited (e.g. Rotman et al. 2012; Roy et al. 2012) and can be considered a typical classification. The levels of involvement align to large extend with the governance structures defined by Conrad's and Hilchey's (2011): consultative, collaborative and transformative governance. In transformative governance citizens initiate project. An example can be found in the global community monitor that measures air and water quality on a global scale (Conrad and Hilchey, 2011).

For the purposes of public education, raising awareness, improve management and policy development, science communication literature (e.g Varner, 2014) provides a more useful classification of the interaction between citizens and professional counterparts. We use the term science communication here in a broad sense and include communication between public and all professional counterparts referred to by Silvertown (2009) including practitioners. Citizen science is often viewed as a form of informal science education, contributing to public awareness of science (PAS) and public understanding of science (PUS). We think this view is too limited, as it only encompasses the deficit model of science communication. Higher levels of involvement public engagement of science (PES) and public participation in science (PPS) are possible (Van der Auweraert, 2005), particularly in the collaborative and co-created projects in Bonney's definition. An example of such participation is the water quality program on Rhode island mentioned in the review by Conrad and Hilchey (2011).

The motivation of professional counterparts to engage in citizen science has been less frequently studied than the motivation of citizens and to our best knowledge research on professionals' motivation is limited to scientists. Scientists' motivations are primarily to advance science as well as develop their careers (Rotman et al., 2012). This is aligned with citizens' motivational desire to contribute to science and conservation or to engage in exploring a topic of their interest further (e.g. Rotman et al., 2012). Weng (2015) identifies three areas of friction between the vision of scientists and volunteers with

regard to citizen science. The first area of friction is often short-term participation of volunteers that conflicts with scientists' interest in long-term processes. The second area concerns the limits of what volunteers can do and their dissatisfaction with the research processes. The third area regards a power hierarchy between citizens and scientists. Rotman et al. (2012) found that while the motivations of citizens and scientists are complementary, they can also change over time. Therefore, continued

attention by those who are managing citizen science projects with regard to matching these motivations is crucial.

The motivation of scientists cannot be translated one-to-one to practitioners or (local) government representatives for two reasons. First, scientists are concerned with scientific data collection (Rotman et al., 2012), while practitioners are often interested in improving management practices (Weng, 2015) and government agencies are concerned with policy making

(Hollow et al., 2015). Second, the different role of authorities leads to different expectations. Water authorities believe that citizens see water resource management as a task for authorities only, which implies that citizen do not want to be involved. Nevertheless, most water authorities agree that they need the observations of citizens for their work as expressed in several studies related to flood risk management (Wehn and Evers, 2014; Wehn, Rusca, Evers, and Lanfranchi, 2015).

This study aims to explore perspectives of government practitioners regarding citizen science. We explore Dutch water practitioners' perceptions on citizen motivation, acceptance in their organisation, (potential) purposes and level of citizen engagement. We address apparent knowledge gaps regarding the motivation of professionals to embrace citizen science and specifically the link with citizen participation. Perceptions were explored using Q-methodology. With Q-methodology we identified a set of viewpoints that describe the variation in opinions. This variation, we consider meaningful for the design

and implementation of citizen science at water boards in the Netherlands. In addition to providing insight in the viewpoints of Dutch practitioners, this study aims to develop a methodology to study perspectives regarding citizen science in a wider range of countries and professionals.

**2 Method**

The study uses Q methodology (Exel and Graaf, 2005; Watts and Stenner, 2012) to find viewpoints on citizen science among

employees of the Dutch water authorities. Q methodology is a relatively uncommon method in water resource management (e.g. Raadgever et al., 2008), but it is a popular method in social sciences fields, such as political science and psychology (Cools et al., 2009). The strengths of the Q methodology are that it combines qualitative and quantitative aspects and that it is statistically robust with small samples of 30-40 people (Exel and Graaf, 2005; Watts and Stenner, 2012). For this research Q methodology has the advantage over fully quantitative approaches in that we could explore a wide spectrum of viewpoints

whereas quantitative methods would arrive easily at averaged values. Quantitative methods would likely not reveal the distinguishing elements that are in our opinion crucial to be aware of for effective implementation of citizen science. Q

methodology has the advantage over fully qualitative methods that it reduces the variation in opinions to a representative small set of viewpoints in a trackable, relatively little biased way (further discussed below).

This section provides a short description of the methodology and research specific details. We kindly refer readers interested in more methodological details to Van Exel and De Graaf (2005) for a quick introduction, Watts and Stenner (2012) or Brown (1980) for an elaboration on the philosophy and statistical base. In conducting this research we closely followed the guidelines of Watts and Stenner (2012). Our Q methodological research consisted of seven stages summarised in Figure 1 and elaborated below.

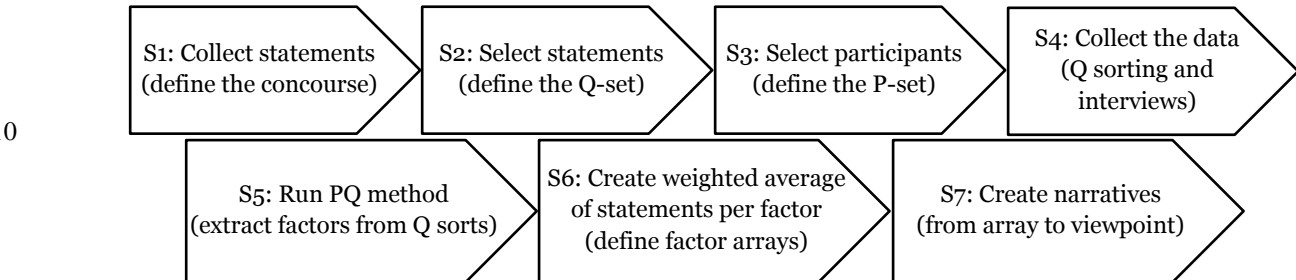

**Figure 1: Flowchart of the steps of the Q methodological research approach. Based on Watts and Stenner (2012). For clarity reasons, we choose to describe Stage 5 in Watts and Stenner (2012) as three separate stages.**

The first stage aimed at collecting an as widely as possible range of opinions on the topic of citizen science for water quality monitoring (discourse) and document them in statements (concourse). We held ten semi-structured interviews with employees of water authorities, nature managers and citizen organizations from which 181 statements were derived. Additionally, we organised a structured group discussion about citizen science with water professionals at the before-mentioned workshop which resulted in an additional 21 statements. To collect a wider range of opinions, we organised a focus group meeting with five middle-aged woman of an informal walking club with high potential to participate in citizen science which resulted in an additional 20 statements. Finally, benefits and downsides of citizen science were extracted from literature which resulted in an additional 7 statements. To reduce researcher bias, we based statements as much and as closely as possible on quotes. The final concourse (all possible statements on a topic) consisted of 229 statements. Four themes could be identified in the concourse: (I) citizen motivation, (II) acceptance of citizen science at the water authority, (III) purposes of citizen science for professionals and (IV) level of citizen engagement.

In Stage 2 the concourse was reduced to the so-called Q-set set of 48 statements that still reflected the full discourse. We first reduced the concourse to a preliminary Q-set of 65 statements by excluding or reframing statements that were similar to others, out of scope or ambiguous. Six master students between the age of 22 and 25 tested this preliminary Q-set. Two

female students had a major in water resource management, the other two female and two male students had a different major (medicine, mathematics, mechanical engineering and management studies). We instructed them to sort the statements and list statements that they did not understand, found similar in meaning or considered irrelevant and improved the preliminary Q-set based on their feedback. Table 1 contains the final Q-set of 46 statements, the roman numbers indicate to what theme the statement contributes.

In Stage 3, the P-set (i.e. the group of participants) was sampled using both a structured approach with three criteria and snowball sampling. Flood risk was a first criterion, as water authorities with a high flood risk face different challenges than water authorities with a lower flood risk. Age (expressed in years since the last reform or merger with another water authority) was a second criterion, as a recent reform suggests the organisation may be more susceptible to innovation, such as citizen science. Location (within or outside the urban conglomerate Randstad) was a third criterion, because interviewees in the semi-structured interviews held to define the discourse suggested a different relation between water authority and population in rural and urban areas. The structured approach resulted in eight water authorities representing a mixture of these criteria. In each water authority we approached the ecologists because we expected them ecologists to be more familiar with the concept of citizen science and we had access to a list of ecologists per water authority. Additionally we used snowball sampling. We asked the participating ecologists to recruit colleagues with a similar opinion, to enhance overlap between individual opinions, and with a different opinions, to increase diversity of opinions. Also, we asked participants to recommend someone with an opposing opinion, in order to discover as many viewpoints as possible. Participants #20, #24, #25, #30 and #31 out of 33 were recruited with this strategy, with the aim that they would belong to different (new) viewpoints. Two to six people with different positions were interviewed per water authority, which resulted in interviews with one politician, twenty policy advisors, ten ecologists/hydrologists and two field staff members.

Next, Stage 4,the Q-sorts (the actual arranging sorting process) took place in four sub-steps, taking a total time of 60 to 75 minutes. First, the first author gave three examples of consultative citizen science to all participants, to ensure everyone had a basic level of understanding of citizen science. These examples were:

- the Dutch garden bird count (www.tuintelling.nl);
- iSPEX, citizens measured particulate matter with a smartphone device called iSPEX (Snik et al., 2014, p. 7351);
- water level monitoring by citizens in a Dutch water authority (UvW, 2015b, p. 15).

Second, participants pre-sorted the statements in three piles: agree, disagree and neutral. Third, they made a final sorting of the statements in a fixed distribution (see Figure 2). Finally, the first author held a structured post-sorting interview. Post-sorting interviews were included in this study, because they can provide in-depth insight in to the beliefs and values underlying the sorts and allow for an analysis based on the participants' rationale rather than on the available literature or the researcher's bias (Gallagher and Porock, 2010). Discussing all statements with participants would have been preferable, but was not feasible given the available time for this study and the geographic spreading of participants. In a structured

interview, participants explained their reasoning for the statements in categories +4 and -4 and (if time allowed) any statement of their choice. Participant's afterthoughts were recorded, transcribed and categorised per statement and per factor.

| Disagree | | | | | | | | Agree |
|---|---|---|---|---|---|---|---|---|
| **-4** (2) | **-3** (3) | **-2** (5) | **-1** (8) | **0** (10) | **+1** (8) | **+2** (5) | **+3** (3) | **+4** (2) |

**Figure 2: The fixed distribution used in this study. The participant places the two statements that he agrees most with in the +4 column, the next three statements in the +3 column, etc. The process is repeated for the disagree statements.**

Next, Stage 5, a factor analysis was performed with the software package PQMethod, version 3.2.1. PQMethod is used to perform a factor analysis and frequently used in Q methodological research (Van Exel and De Graaf, 2005; Cools et al., 2009; Raadgever et al., 2008; Watts and Stenner, 2012). A factor analysis is a statistical method to describe variability in a set of correlated variables, in this case the ranking of statements by individuals, by a smaller number of factors. We included three factors with an eigenvalue above 1(recommended by Watts and Stenner, 2012) for further analysis. The factor analysis also provided which people load on which factor, i.e. which people have a perspective resembling that factor. People can load on none, one or multiple factors. For this a threshold was used, the Significant Factor Loading (SFL). In this study we followed Watts and Stenner (2012) and used a SFL of 0.38. Factors were optimized using factor rotation, which aims to have as many people load on a single factor, rather than loading on two factors simultaneously. The rotating process does not alter the results themselves, but changes the researcher's observation position in order to optimise the loading of each Q-sort on a single factor (Watts and Stenner, 2012, p. 118). A manual rotation was preferred above the built-in Varimax rotation of the PQMethod, because it has a lower inter-factor correlation and thus results in more distinct viewpoints.

Next, Stage 6, a weighted average was used to create an illusory person with a factor loading of 1.0 per factor, i.e. a hypothetical person who has fully adopted this factor. Following Watts and Steiner (2012) Q-sorts with loadings that exceed the Significant Factor Loading (SFL) of 0.38 for a factor were incorporated to compute the weighted average for that factor. See Table 2 for the factor arrays and Table 3 for the final factor loadings. It must be noted that the number of people loading

on a factor cannot be used to determine the distribution of viewpoints in the total population without additional (quantitative) research.

The final stage, Stage 7, is data analysis, where the factor arrays were translated into a viewpoint narrative. We followed the guidelines of Watts and Stenner (2012), Gallagher and Porock (2010) and Cools et al. (2009) and used *distinguishing statements*. We created a narrative of the +4 and -4 ranked statements and the statements ranked highest in a single factor array, meaning this statement is ranked lower in all other factors, and vice versa. For example, in Factor A the statements 2 and 9 ranked with +4, Statements 10 and 41 with -4 and Statement 34 is an example of a Statement ranked highest in factor A. Factor A ranks it +2, compared to -1 and +1 in factors B and C. Hence, Statements 2, 9, 10, 41 and 34 were included the viewpoint narrative of Factor A. In addition to this mechanism of distinguishing statements, two other mechanisms were introduced to reduce researcher bias. First, we conducted post-sorting interviews in order to be able to incorporate participant's underlying values and assumptions in the process of interpretation. During these interviews we also checked if in their opinion all relevant aspects were covered by the Qset. Second, we showed all participants an initial version of the narratives and asked whether they recognised themselves in their assigned narrative viewpoint and why (or why not). In addition we presented the results to employees of the Dutch water authorities at two occasions (Delfland Scriptieprijs uitreiking 21-1-2016 and STOWA Monitoringcongres 19-4-2016) to collect feedback. An overview of mechanisms used to reduce bias in all steps of the research is given in Figure 3.

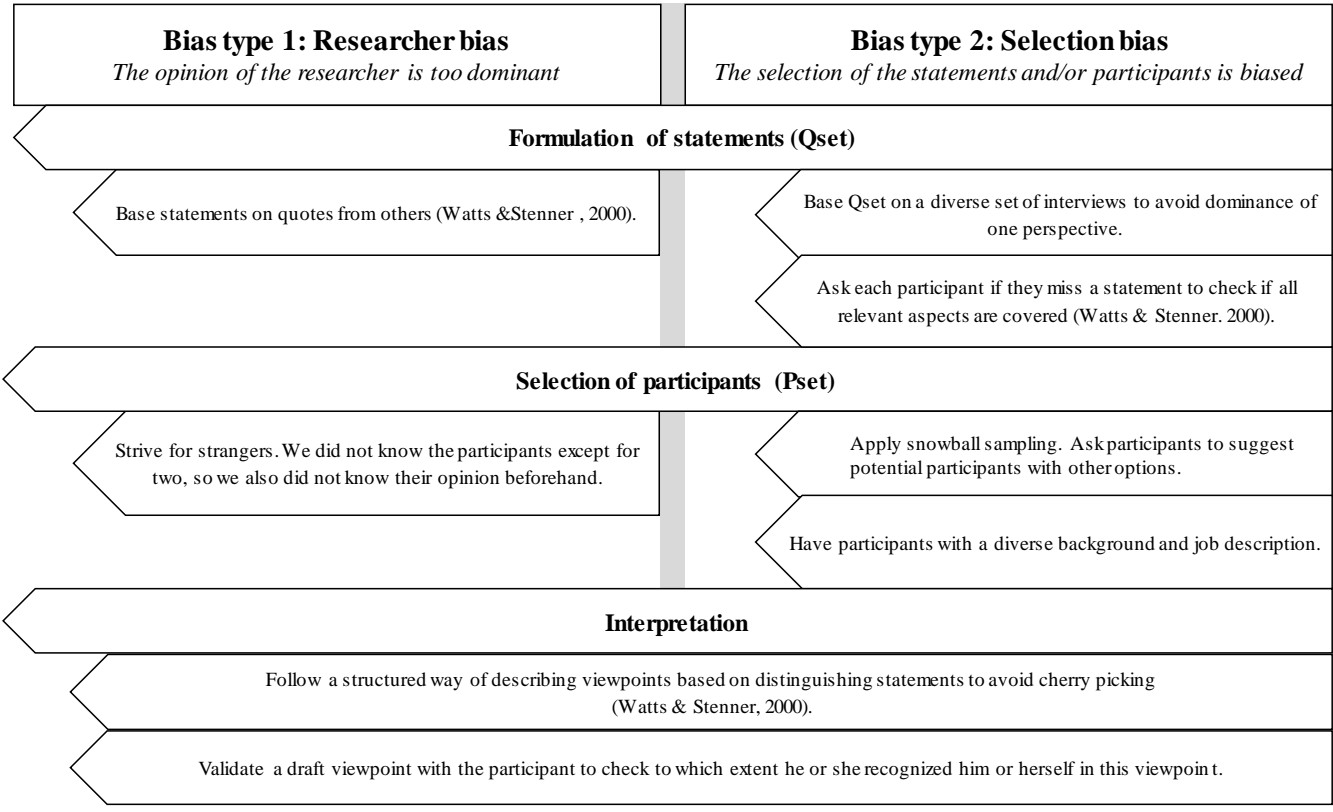

| Bias type 1: Researcher bias | Bias type 2: Selection bias |
|---|---|
| *The opinion of the researcher is too dominant* | *The selection of the statements and/or participants is biased* |

**Formulation of statements (Qset)**

Base statements on quotes from others (Watts &Stenner , 2000).

Base Qset on a diverse set of interviews to avoid dominance of one perspective.

Ask each participant if they miss a statement to check if all relevant aspects are covered (Watts & Stenner. 2000).

**Selection of participants (Pset)**

Strive for strangers. We did not know the participants except for two, so we also did not know their opinion beforehand.

Apply snowball sampling. Ask participants to suggest potential participants with other options.

Have participants with a diverse background and job description.

**Interpretation**

Follow a structured way of describing viewpoints based on distinguishing statements to avoid cherry picking (Watts & Stenner, 2000).

Validate a draft viewpoint with the participant to check to which extent he or she recognized him or herself in this viewpoint.

**Figure 3: A summary of the mechanisms used to reduce bias. In two columns actions are listed to reduce researcher bias (left) and selection bias (right).**

## 3 Results

The 33 Q-sorts resulted in the identification of three factors from which three viewpoints were derived: A "Citizen participation for data application", B "Water authority in control", C "Education and sharing local knowledge". The choice for three factors was based on the explained variance of the first four factors before rotation that displayed with 53%, 8%, 6% and 1% a cut off after the third factor that was supported by qualitative information from the interviews. Table 1 and the radar charts presented in Figure 5 show how an individual would rank the items if that person was representing that factor 100%. For example, statement 9 ("Citizen Science enables the collection of large amounts of measurements", Theme III Purposes of Citizen Science) would be placed in the most agree (column +4) by a person with Factor A, under agree (column +2) for Factor B and in the neutral (column 0) for Factor C. None of the viewpoints disagrees with Statement 9, but the difference between Viewpoint A and C is evident. Except for Statement 2 ("Citizen Science is important, since it contributes to increasing water awareness"), to which all viewpoints fully agree, and Statement 35("In citizens are structurally

contributing they should be compensated for that"), differences were found between the viewpoints that will be further discussed below. The factors provided a quite clear separation of the participating practitioners in groups given Table 2. Out of 33 participants, 21 loaded significantly and uniquely on Factor A, 4 on Factor B and 2 on Factor C. Three participants loaded significantly on Factor A and C, one on Factor A and B and one on Factor B and C. One participant did not load significantly on any of the factors.

The remainder of this section contains the three viewpoint narratives. The term viewpoint is used to refer to the factor's interpretation for which we made use of the quantitative factor arrays in Table 1 and qualitative quotes from participants loading significantly on that factor (see Table 2). The narratives are based on absolute results (agree or disagree), the relative results (an item is ranked higher or lower in viewpoint A than B and C) and characteristic interview items. Item rankings are presented in the following format: (*item number : item ranking*) such that (2: +4) means item #02 is ranked +4 in this viewpoint. Interview fragments are integrated in the narratives as a quote followed by the letter Q and a number indicating the source. For example ("quote" – Q1) means the quote comes from the Q-sort and thus participant 1. Figure 4 shows the availability of interview fragments per factor and per statement. Most fragments (125) were available to interpret Factor A as most people loaded significantly to that Factor A and less for Factor B (20 fragments) and C (10 fragments). As an intended result of the interview technique (see Stage 4 in the Method Section) most fragments were available for statements with particularly low or high rankings and there was a positive relation between the level of disagreement among factors and the number of interview fragments. The higher the total absolute difference between the Factors A, B and C, the higher the number of interview fragments.

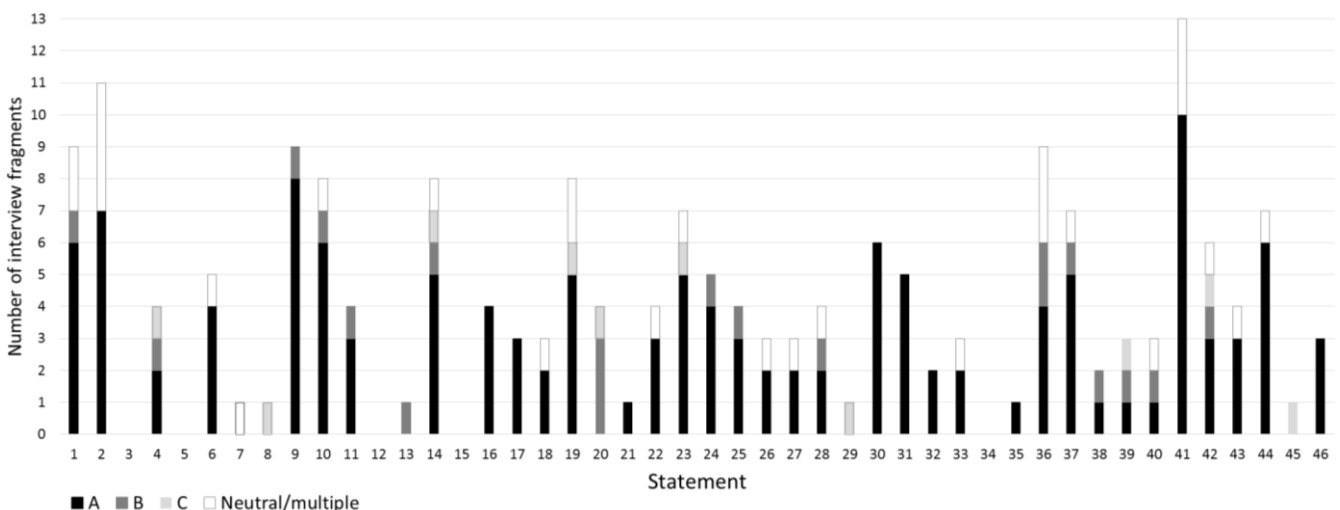

**Figure 4: Distribution of interview fragments available per viewpoints and per statement. Interview fragments of people that load to no or two viewpoints were categorized as 'neutral'.**

**Table 1: Final factor arrays, the numbers in columns A, B and C are the theoretical item score for a person whose viewpoint is 100% that factor. Roman numbers indicate the theme category (cat) of the item. I = citizen motivation, II = water authority acceptance of citizen science, III = purposes, IV = level of engagement.**

| | Statements (QSet) | Cat | A | B | C |
|---|---|---|---|---|---|
| 1 | Providing citizens with insight in water quality will only lead to unnecessary panic and questions. | II | -3 | -4 | -4 |
| 2 | Citizen Science is important, since it contributes to increasing water awareness. | III | +4 | +4 | +4 |
| 3 | Citizen Science is a solution to explain why you take certain measures as a water authority. | III | +1 | -1 | -1 |
| 4 | Water quality is an abstract concept, citizens will not understand what they measure. | IV | -1 | -2 | -3 |
| 5 | It is important to have proper communications to citizens about why values deviate from the norm and what the uncertainty in the measured value is. | III | +1 | 0 | +1 |
| 6 | I would not know why citizens would not be interested in monitoring water quality. | I | -1 | -2 | 0 |
| 7 | Citizen Science is an economical way to collect (extra) measurements. | II | +1 | +1 | -1 |
| 8 | Citizen Science enables the collection of more measurements by conducting them more frequently. | III | +3 | +3 | +1 |
| 9 | Citizen Science enables the collection of large amounts of measurements. | III | +4 | +2 | 0 |
| 10 | Measurements and observations by citizens are no valuable addition to the official monitoring network. | III | -4 | -2 | -1 |
| 11 | The most important goal is that the measurement data provide value to the water authority because the organisation has invested its time and energy. | III | 0 | +1 | +1 |
| 12 | I would rather make (smart) use of existing measurements than let citizens' conduct more measurements. | II | -1 | 0 | 0 |
| 13 | The greatest challenge is how to teach people something, if they can or want to spend little time on it. | III | 0 | 0 | -1 |
| 14 | Schools are especially suitable target groups to conduct these measurements, for example during a 'water lesson'. | III | 0 | 0 | 2 |
| 15 | The most important goal of citizen science is to teach people something about the environment they live in. | III | +1 | +2 | +3 |
| 16 | Citizen Science is an interesting social innovation, but not suitable for actually collecting useful data. | II | -2 | -2 | 0 |
| 17 | Citizens' abilities are often under estimated; they are better educated and smarter than we think. | II | +1 | +1 | 0 |

| | Item | cat | A | B | C |
|---|---|---|---|---|---|
| 18 | As a water authority we need to learn how to handle the uncertainty of alternative (cheap) measurements that originate from Citizen Science. | II | +2 | +1 | +1 |
| 19 | Data collection by citizens is unreliable and should not be accepted by the water authority. | III | -3 | -2 | -1 |
| 20 | Citizens will only participate in Citizen Science, if participation is in their own interest. | I | 0 | +2 | -2 |
| 21 | Not all citizens can be trusted to conduct these measurements. | II | -1 | +1 | 0 |
| 22 | With a short training, citizens will be able to conduct measurements for the water authority. | IV | 2 | +1 | 2 |
| 23 | Citizen Science is an interesting way to give meaning to the concept of citizen participation. | III | +3 | +1 | +3 |
| 24 | Citizen Science is necessary, because it helps to decrease the awareness gap between citizens and the water authority. | III | +2 | -3 | +2 |
| 25 | By using Citizen Science, the water authority shows that it is keeping pace with the times. | II | +1 | -1 | 0 |
| 26 | An important advantage of Citizen Science is that it reduces citizen's resistance to projects. | III | 0 | 0 | +1 |
| 27 | One can connect with and involve another part of the audience using Citizen Science. | III | +2 | 0 | +3 |
| 28 | As long as Citizen Science is not included in the policy at the top levels, the water authority should not invest in it. | II | -3 | -3 | -2 |
| 29 | It is a major bottleneck to create support within the water authority for the deployment of Citizen Science. | II | -1 | 0 | -1 |
| 30 | The water authority will benefit from using Citizen Science in conducting its tasks, because less (financial) resources are available. | II | 0 | -1 | 0 |
| 31 | The conservative character of my organisation is a major bottleneck for Citizen Science. | II | -1 | -1 | -2 |
| 32 | The organisation is not equipped to work with large groups of citizen scientists. | II | 0 | +3 | 0 |
| 33 | My organisation has no capacity to work with all this data. | II | -2 | -1 | -2 |
| 34 | The water authority should incorporate in its policy how to deploy and stimulate Citizen Science more. | II | +2 | -1 | +1 |
| 35 | If citizens are structurally contributing, they should be compensated for that. | I | 0 | 0 | 0 |
| 36 | If citizens collect data for the water authority, they should have a say in the measures taken afterwards. | III | -2 | -4 | -3 |

| | Item | cat | A | B | C |
|---|---|---|---|---|---|
| 37 | Citizens often have local knowledge and the water authority should use this knowledge. | IV | +3 | +4 | +4 |
| 38 | Citizen Science is important, because it gives insight into the problems that citizens are concerned with. | IV | +1 | 0 | +1 |
| 39 | Citizens should have insight in the most recent information of the water quality that is available with the water authority. | IV | +1 | +1 | +2 |
| 40 | If you provide citizens with a reference framework, they themselves can validate their data. | IV | 0 | -3 | -3 |
| 41 | I do not want citizens to interfere with our work. | II | -4 | -1 | -4 |
| 42 | The water authority should maintain control of conducting measurements, since the water authority is indeed responsible. | IV | -2 | +3 | +2 |
| 43 | I think the creation of Citizen Science does not fall within the tasks of the water authority. | II | -2 | -1 | -2 |
| 44 | I do not have a full image of what is possible with Citizen Science. | II | -1 | 0 | -1 |
| 45 | An important caveat is that citizens will expect that their measurements will have a direct influence on policy. | III | 0 | +2 | +1 |
| 46 | Citizens cannot be motivated to participate in such projects for a long period. | I | -1 | +2 | -1 |

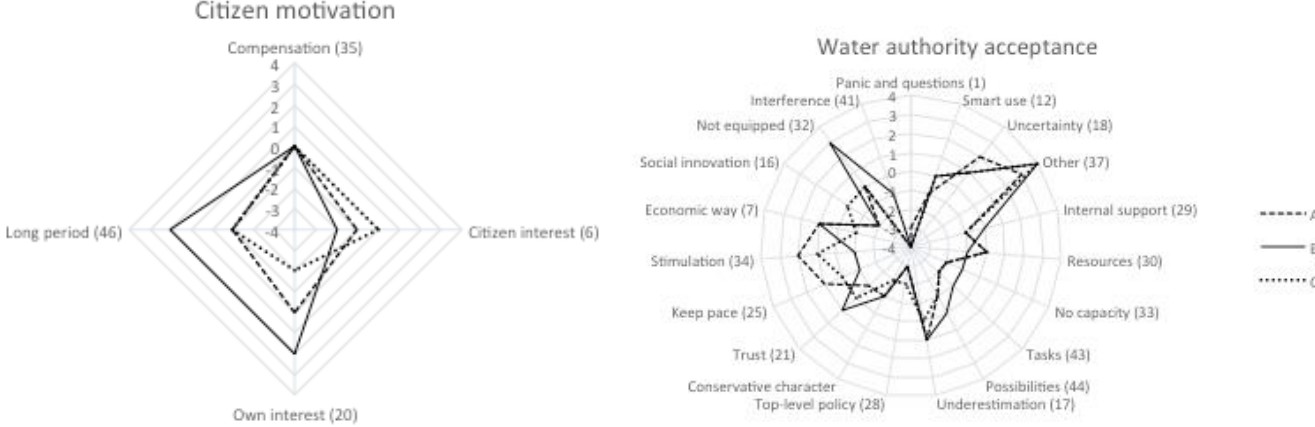

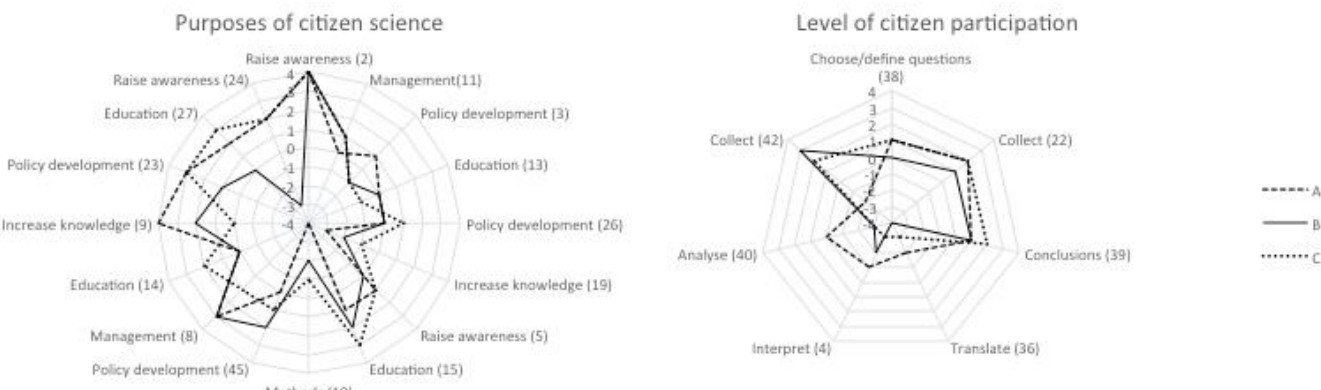

**Figure 5: Theoretical item score for a person whose viewpoint is 100% Factor A, B or C. Statements are clustered per theme (Citizen motivation, water authority acceptance, purposes and level) in four radar charts. Numbers in brackets refer to the full statement given in Table 1.**

**Table 2: Final factor loadings after rotation. Sorts in bold indicate that a person's loading exceeds the significant factor loading (SFL) and a person's viewpoint thus resembles viewpoint A, B or C. Also included are characteristics of respondents: water authority (Identification (ID), elevation above sea level (El), located in Randstad (RS) and recently reformed (since 2005)) and individual characteristics (sex, age and function group).**

| sort | Factors | | | Water authority | | | | Individual respondent | | |
|---|---|---|---|---|---|---|---|---|---|---|
| | **A** | **B** | **C** | ID | El. | RS | RR | **Sex** | **Age** | **Function group** |
| 1 | **0,74** | 0,18 | -0,01 | A | - | Yes | Yes | F | 45-65 | Politician |
| 2 | **0,49** | 0,18 | 0,35 | A | - | Yes | Yes | M | 20-45 | Policy advisor |
| 3 | 0,35 | **0,55** | 0,18 | A | - | Yes | Yes | M | 45-65 | Policy advisor |
| 4 | **0,49** | 0,36 | -0,03 | B | - | Yes | No | F | 20-45 | Advisor (water quality/ecology) |
| 5 | **0,69** | -0,07 | 0,29 | C | + | No | Yes | F | 20-45 | Policy advisor |
| 6 | **0,71** | 0,09 | 0,32 | C | - | Yes | No | M | 20-45 | Policy advisor |
| 7 | **0,45** | **0,62** | 0,12 | D | - | Yes | No | M | 45-65 | Field staff |
| 8 | 0,31 | **0,46** | **0,43** | D | - | Yes | No | F | 20-45 | Advisor (ecological monitoring) |
| 9 | **0,79** | 0,19 | 0,06 | D | - | Yes | No | F | 20-45 | Policy advisor |
| 10 | -0,05 | 0,22 | **0,66** | D | - | Yes | No | M | 45-65 | Advisor (ecology) |
| 11 | **0,64** | -0,17 | 0,03 | C | + | No | Yes | M | 45-65 | Policy advisor |
| 12 | **0,72** | 0,12 | **0,49** | C | + | No | Yes | M | 45-65 | Policy advisor |
| 13 | **0,75** | 0,01 | -0,12 | C | + | No | Yes | M | 20-45 | Advisor (innovation) |
| 14 | **0,69** | -0,06 | **0,41** | C | + | No | Yes | F | 45-65 | Field staff |
| 15 | **0,84** | -0,06 | 0,30 | E | + | No | Yes | F | 45-65 | Policy advisor |
| 16 | **0,64** | 0,33 | **0,42** | E | + | No | Yes | F | 20-45 | Advisor (permits and assessment) |
| 17 | **0,58** | -0,05 | 0,25 | E | + | No | Yes | M | 45-65 | Advisor (water system) |
| 18 | -0,04 | **0,55** | 0,24 | F | - | No | Yes | M | 45-65 | Policy advisor |
| 19 | **0,69** | 0,01 | 0,17 | F | - | No | Yes | M | 20-45 | Hydrologist |
| 20 | 0,32 | **0,43** | -0,13 | F | - | No | Yes | M | 45-65 | Policy advisor |
| 21 | **0,73** | -0,09 | 0,32 | F | - | No | Yes | M | 45-65 | Policy advisor |

| | | | | | | | | | | |
|---|---|---|---|---|---|---|---|---|---|---|
| 22 | **0,59** | 0,25 | 0,04 | F | - | No | Yes | F | 20-45 | Hydrologist |
| 23 | **0,82** | 0,17 | 0,01 | G | - | Yes | Yes | F | 20-45 | Advisor (innovation) |
| 24 | **0,74** | 0,04 | -0,12 | G | - | Yes | Yes | M | 20-45 | Advisor (innovation) |
| 25 | **0,45** | 0,00 | 0,23 | G | - | Yes | Yes | M | 20-45 | Ecologist |
| 26 | **0,80** | -0,06 | -0,07 | G | - | Yes | Yes | M | 45-65 | Hydrologist |
| 27 | **0,85** | 0,15 | 0,23 | G | - | Yes | Yes | M | 45-65 | Policy advisor |
| 28 | **0,54** | 0,20 | 0,27 | H | - | Yes | No | M | 45-65 | Policy advisor |
| 29 | 0,34 | -0,03 | 0,34 | H | - | Yes | No | M | 45-65 | Project leader |
| 30 | 0,30 | **0,39** | -0,09 | A | - | Yes | Yes | M | 20-45 | Policy advisor |
| 31 | 0,27 | 0,23 | **0,43** | B | - | Yes | No | M | 45-65 | Policy advisor |
| 32 | **0,77** | 0,21 | 0,10 | B | - | Yes | No | F | 20-45 | Advisor (water quality) |
| 33 | **0,79** | -0,22 | 0,34 | B | - | Yes | No | F | 20-45 | Policy advisor |
| EV[1] | 12.55 | 2.29 | 2.58 | | | | | | | |
| Var[2] | 38% | 7% | 8% | | | | | | | |

### 3.1 Viewpoint A: "Citizen participation for data application"

The people loading on Factor A (see Table 2) are a mixture of hydrologists, advisors, policy advisors, field staff and a politician. In this group are fourteen men and eleven women. Eleven people are middle aged (between 45 and 65). They are representing all eight incorporated water authorities, which are located within and outside the Randstad and have a mixture of higher and lower flood risk. Nineteen people work at a water authority that has recently gone through an organizational reform.

People with Viewpoint A think that citizen science is important for water authorities to increase water awareness (2: +4), because citizens are unacquainted with the work of the water authority *"People often do not know what the water authority is doing exactly and we do not really stand out. Citizens sometimes really wonder what they pay tax for [...]"* (Q5) and *"what they can do themselves to improve water quality."* (Q27) . They believe that their water authority should actively incorporate citizen science in its policy (34: +2) and that the water authority should not wait to invest in citizen science until it is included in top-level policies (28: -3).

Additionally, people with this viewpoint value citizen science for the collection of large amounts of data (9: +4) and for conducting measurements more frequently (8: +3). *"This data, they are an opportunity to have an area covering insight in dynamics of water quality and ecology."* – Q26. They think that citizens can be trusted to conduct these measurements (21: -1). Although citizen science is a social innovation and the acquired data are less accurate, it should be accepted by the water authority (19: -3) (*"I mainly disagree strongly with the latter part of this statement."* – Q13 *[(...) and should not be accepted by the water authority]*) and will be a valuable addition to the official monitoring network (10: -4). These people do not prefer the smart use of existing data to citizen science data (12: -1). The organisation is expected to have sufficient capacity to analyse all the data (33: -2) at the moment, but the water authority has to learn how to handle the uncertainty of these alternative (often more economical) measurements (18: +2). They do not believe that the water authority needs to maintain control of monitoring, even though water authorities are in the end responsible for monitoring (42: -2). *"This is nonsense, because a lot is already measured by other parties."* – Q25. They believe that citizens, if provided with a reference framework, can validate their own data (40: 0)*"If they know what to do with it [the results], they can translate it to their environment."* – Q11.

People with Viewpoint A think of citizen science as an interesting way to give meaning to the concept of citizen participation (23: +3) and decrease the gap between the water authority and citizens (24: +2). They are least (compared to Viewpoint B and C) afraid that citizens will expect their contribution to have a direct impact on policy (45: 0) and they do not think citizens should get this influence (36: -2). *"You should prevent that, because manipulation [of results] is evident."* – Q9. They consider citizen science to be a solution when it comes to explaining why you undertake certain measures (3:

+1). This group further feels that citizen science will show that the water authority is keeping pace with the times (25: +1), although it is not a priority. Giving citizens' insight in water quality will not lead to unnecessary questions and panic (1: -3). *"Those questions will come, but you should not be afraid, not afraid to say that you do not know everything."* – Q21. They do not fear citizen interference with their work (41: -4).

### 3.2 Viewpoint B: "The water authority in control"

The six people loading significantly on Factor B (see Table 2) form a mixture of advisors, policy advisors and field staff. Five out of six are male and four of them are middle aged. They work at three different water authorities, two people work outside the Randstad. Four people work at water authorities that have recently (after 2005) gone through an organizational reform. All work in an area with a high flood risk.

People with Viewpoint B consider citizen science important for increasing water awareness (2: +4). In contrast to Viewpoint A, they think that the water authority should not incorporate citizen science as part of its policy (34: -1). They feel they not have a full idea (yet) of what is possible with citizen science (44:0). However they do think the water authority should invest in citizen science, even if it is not yet included in top level policies (28: -3). They do fear that citizens cannot be motivated for a long-term participation (46: +2) and will not participate unless participation is in their own interest (20: +2). This group is more concerned than the other two groups that the creation of a support base within the organisation will be a bottleneck (29: 0) and they are convinced that their organisation is not (yet) equipped to work with large groups of citizen scientists (32: +3).

People with Viewpoint B believe that local knowledge will be valuable for the water authority (37: +4), as the citizen *"knows his own environment better than we do, on the small scale. We only have the broad overview in a large area"* – Q20. They strongly believe that the water authority needs to maintain control of monitoring, because it has the final responsibility (42: +3). *"I know what should be done with the data in the end. If we leave it to volunteers in this case, you have no reassurance on what comes when."* – Q3. Citizen Science allows for the collection of more measurements (9: +2) by conducting them more frequently (8: +3). They think citizens will be able to conduct measurements after they receive a short training (22: +1), but they do not expect that citizens will be able to validate their own data if provided with a reference framework (40: -3). *"If [data] quality is important to you, I am not sure whether citizens can do this."* – Q18. They further question whether all citizens can be trusted in doing measurements (21: +1). *"If the citizen does not have personal interest, you have to wait to see what happens. Then he will think: I do not feel like it, I do not have time."* – Q20. This reflects their belief that most citizens would not be interested in participating (6: -2). This group does not believe the water authority needs citizen science to help fulfil its tasks to compensate for less financial resources (30: -1).

People with this viewpoint are least convinced that citizen science will involve another part of the public (27: 0) or that it is an interesting way to give meaning to citizen participation (23: 1). Moreover, they believe that citizen science should not be used to decrease the gap between citizens and water authorities (24: -3) or to show that the water authority is keeping pace with the times (25: -1). *"If this is your reason, I think it is rather cheap."* – Q18. If citizens start collecting data for the water authority, this group strongly feels that they should not be given more influence over measures (36: -4), but they do fear that citizens will think that their work will influence policy directly (45: +2). *"Citizens, I would almost say per definition, cannot do that [balance interests], they just want to do what they want."* – Q30. People in this group do not fear questions or panic from citizens (1: -4). *"If you are so suspicious towards your citizens, you have to question your role as government."* – Q3.

### 3.3. Viewpoint C: "Education and sharing local knowledge"

The six people significantly loading on Factor C (see Table 2) are a mixture of advisors, policy advisors and field staff. Four people are middle aged and three of them are male. They work at four different water authorities, three respondents within and three outside the Randstad. Three people work at a water authority that has recently (after 2005) gone through an organisational reform. Three out of six people with Viewpoint C work in an area with a lower flood risk.

People in Viewpoint C think that citizen science is important, because it contributes to increase of water awareness (2: +4). The most important purpose for this group is to teach people something about their environment (15: +3) and especially schools are considered to be a good target audience (14: +2). *"It is a good way to keep them [students] engaged."* – Q10. They think that citizens will understand what they measure, even though water quality is an abstract concept (4: -3). *"I think this is an offensive comment toward the citizens, as if they are stupid."* – Q10. They believe it is possible to teach people something within a short period of time (13: -1) and they find it difficult to think of reasons why people would not be interested in water quality (6: 0) compared to the other two groups. In contrast to Viewpoint B, they do think that citizens will participate, even if participation does not directly serve their own interests (20: -2). *"I participate as a citizen in a sort of science project, I do not do that for my own benefit, but because I like it and want to contribute. I think I am not the only one"* – Q10. Also, the conservative character of water authorities is not considered a major bottleneck (31: -2).

They feel that the water authority should use the local knowledge that citizens have (37: +4), but they consider citizen science to be merely a social innovation, rather than a way to collect useful data, compared to the other viewpoints (16: 0). This is reflected in their relatively small support of the idea that citizen science will allow for collecting large amounts of data (9: 0) and for conducting measurements more frequently (8: +1). *"It is mainly supportive material and not a replacement of existing sources, because it is invalidated and uncertified information. I do not think that will fit"* – Q31. They strongly reject the view that citizens should not interfere with their work (41: -4), although they believe the water authority should stay in control (42: +2). *"In my opinion information is essential for policy to be good, [so] I think they*

*should be collected by a professional."* – Q10. People in this group believe that citizens will not be able to validate their own data (40: -3).

People with Viewpoint C consider citizen science to be a good way to bind and involve another part of the audience (27: +3), to decrease the gap between citizens and water authorities (24: +2) and, to a lesser extent, to reduce citizens' resistance to projects (26: +1). A caveat could be that citizens will expect their measurements to have a direct influence on policy (45: +1), even though they should not be given a say in the measures taken afterwards (36: -3). *"For me these are two separated tracks. [...] they have this influence via the representatives that they can elect for the board."* – Q31. Citizen should be given

insight in the most recent information about water quality that is available with the water authority (39: +2). *"I believe that citizens and everyone have the right to get information from us."* – Q10. These people strongly disagree that providing citizens with insight in water quality will lead to unnecessary panic and questions (1: -4).

## 4 Discussion

This study identified three different viewpoints with regard to citizen science derived with Q methodology. Participants

sorted statements about four themes described in the Introduction and Method section. In this discussion we first (Section 4.1) reflect on these four themes and relate the results to literature and second (Section 4.2) discuss the limitations of the research and makes recommendations for future research.

### 4.1 Reflection on the four themes

The statements contributed to one of the four themes: (I) citizen motivation, (II) acceptance of citizen science at the water

authority, (III) purposes and (IV) level of citizen participation. Several statements related to perceived citizen motivation by the participants (Theme I in Table 1 and upper left radar chart in Figure 5) and acceptance of citizen science at the water authority (Theme II and upper right radar chart in Figure 5). Trust in citizens' motivations and commitment ranged from low in Viewpoint B tot high in Viewpoint C. The assessment of high, low or medium was based of the attitude that emerged from the ranking of statements in Theme I and II. Viewpoint C clearly had the most positive attitude towards citizens' motivations

and B the least positive. For Theme II the distinction was less clear, although all viewpoints express a rather positive attitude. All three viewpoints are rather positive about implementing citizen science (perhaps due to some volunteer bias discussed in Section 4.2), although Viewpoints A and B are concerned with respectively the image of the water authority, and with the organisational capacity and a lack of internal support. Table 3 summarises these findings. Table 4 summarises the results with regard to the support per viewpoint for the purposes listed in the introduction (Theme III). The same has

been done for the roles that citizens can have (Theme IV) in Table 5 according to the classification of Bonney *et al.* (2009).

**Table 3: Summary of the perception of the level of citizen motivations to contribute (Theme 1) and the level of acceptance of citizen science at the water authority (Theme II) for viewpoints A, B and C.**

|  | A | B | C |
|---|---|---|---|
| *Theme 1: Citizen motivations to contribute* | Medium | Low | High |
| *Theme 2: Acceptance of citizen science at the water authority* | High | Medium | High |

**Table 4: Overview of Theme III (purposes for citizen science) as supported in the Viewpoints A, B and C. Purposes clearly indicated are marked with an X. (x) indicates support for this purpose is not convincing.**

|  | Viewpoint A | Viewpoint B | Viewpoint C |
|---|---|---|---|
| *Increase knowledge* | X | X | X |
| *Improve methods* | X |  |  |
| *Raise awareness* | X | X | X |
| *Improve Management* | X | (x) |  |
| *Public education* |  |  | X |
| *Policy development* | X |  | (x) |

All viewpoints embrace data collection by citizens, thus contributory projects, and none support co-created projects (Table 5). Viewpoint A is optimistic towards citizen participation in the analysis of the data (see Statement 40), suggesting a potential for collaborative projects. Viewpoint B and C are wary of involving citizens in these steps of the research process. None of the viewpoints supports statements related to co-created projects. The following explanations are illustrative of the reluctance of participants to involve citizens in topic selection for monitoring (Statement 38): *"There can be [topics] which we think they are important, while citizens do not find it important in the end."* (Q24, Viewpoint A) and *"In that case you should answer all [these questions of citizens] and I think our organisation is not equipped at the moment"* (Q18, Viewpoint B). Regarding the translation of results to action (Statement 36) participants said: *"They [citizens] can only focus on the problems in their direct environment, but not on the implications for a wider area"* (Q19, Viewpoint A); and *"I would not go that far"* (Q18, Viewpoint B). Participants mentioned external causes such as the legal obligations a water authority has regarding the use of standardized methods and reports for water quality monitoring. This is consistent with previous conclusions about the responsibility of the water authority in relation to acceptance of citizen science (Wehn and Evers, 2014). Other participants mentioned internal causes, such as the existing procedures for citizens to influence decision-making, including complaint procedures and the water authority general elections once every four years.

**Table 5: Overview of theme IV (level of citizen participation), based on Bonney et al. (2009). Supported activities are marked with an X. An (x) indicates the activities are sometimes assigned to citizens.**

| | Contributory projects | Collaborative projects | Co-created projects | Viewpoint A | Viewpoint B | Viewpoint C |
|---|---|---|---|---|---|---|
| *Choose or define question(s) for study* | | | X | | | |
| *Design data collection methodologies* | | (x) | X | | | |
| *Collect samples and/or record data* | X | X | X | X | X | X |
| *Analyse samples and data* | (x) | X | X | X | | |
| *Interpret data and raw conclusions* | | (x) | X | | | |
| *Disseminate conclusions/translate results to action* | (x) | (x) | X | | | |
| *Discuss results and ask new questions* | | | X | | | |

There appears to be a mismatch between the intentions of the participants with respect to the purposes that citizen science should support and the expressed level of trust in citizens as reflected by the envisioned participation level. Especially people with Viewpoint C believe citizens have noble motivations and they trust the citizens to a great extent. This trust is not reflected in the roles they envision for citizens, which is limited to data collection. The same goes for Viewpoint A. People with this viewpoint believe citizen science can serve many purposes (see Table 4). However, the envisioned role for citizens is limited to data collection and analysis. A lack of trust in citizens, low intentions to use the citizen scientists' data or a lack of support for higher levels of participation might collide with citizens' motivations as described in the introduction. A relation of mutual trust is required as the basis for effective citizen science projects and prolonged contributions by citizens (Rotman et al., 2012). Viewpoint B reveals distrust in the commitment of citizens, citizens' intentions to participate and their capacity (see Statements 20, 21, 22, 40 and 45). Another important motivation for citizens the provision of feedback on how the data are used (e.g. Bonney et al., 2009; Rotman et al., 2012; Roy et al., 2012) which is particularly in contrast with Viewpoint C. Viewpoint C focuses on the goals of education (see Statements 14 and 15), with little emphasis on the actual use of the data (see Statements 8 and 9). Concerns regarding data quality may be lingered if recent development in systematic modelling and data assimilation approaches (e.g. Clark et al, 2015; Shoaib et al, 2016; Hut et al, 2015) would be adopted at the water authorities in the Netherlands. Such frameworks would allow systematic tracing of the propagation of uncertainty of citizen data in decision support tools and may identify opportunities for citizen science in model structure

determination, which can be the largest source of uncertainty (Shoaib et al, 2016). Ottinger (2010) stressed the need for standardised methods in citizen science and legal embedding to increase the acceptance of the data and actual use of citizen science in policy making. In the Netherlands systematic data handling and standardisation may pave the way towards using citizen science for public participation in line with the vision of the governing body Dutch Water Authorities (UvW, 2015b).

## 4.2 Research limitations and recommendations

Q methodology is an abductive research approach (Watts and Stenner 2012), which means that we tried to understand and explain the data rather than describe it or test a hypothesis. This approach is subjective in nature. Researcher bias was reduced where possible as presented in the framework in Figure 3. By collecting statements from various sources, reasonable saturation in the statements that formed the discourse was achieved and confirmed by that no clearly new statements arose from the post sorting interviews. The second sampling strategy (see Stage 4 in the Method section) recruited five participants that were expected to have different viewpoints. Two of them had Viewpoint B and one had Viewpoint C, thus broadening the scope of viewpoints. Still the voluntary nature of participation might have attracted participants with a positive attitude towards citizen science. At several occasions the research was presented to water authority employees. They were surprised that all viewpoints were relatively positive about citizen science, because they would have believed to have colleagues who are indeed more sceptical than these viewpoints.

To further reduce researcher bias, we asked participants whether they recognised themselves in the assigned viewpoint described in an early draft of the text presented in the Results section. 15 Participants responded, 13 of which fully identified with their viewpoint, because they agreed to the viewpoints' main assertions. Two respondents were in doubt, due to overlap between Viewpoint A and C. The correlation between Factor A and C was 0.43, which indeed indicates that they are interrelated and overlap. Typically correlations above 0.39 are considered significant (Watts and Stenner, 2012). Still Factor A and Factor C are considered sufficiently different to regard them as the basis for separate viewpoints, particularly given the, (in our opinion crucially) different opinion on involving citizens for data collection. Correlations between Factor A and B and Factor B and C were lower and not significant, respectively 0.26 and 0.35. A mechanism to further reduce bias, beyond the mechanisms in Figure 3, could be to execute stages such as statement definition, interviews and transcript analysis by independently working researchers. Yet this was not feasible within the scope of this research.

The importance of post-sort interviews, as stated by Gallagher and Porock (2010), was recognised in this work. Particularly statements with multiple fragments per viewpoint illustrated this. For example, the interview fragments for Statement 36 revealed a difference in reasoning to exclude citizens from decision making on measures to realize policy goals. Participants with Viewpoint A feared manipulation of results, while participants with Viewpoint B emphasised the responsibility of the water authority to take an informed decision and balance conflicting interests. Hence, reformulation and/or splitting of Statement 36 may be considered if the set of statements was adopted for future research. Four participants with Viewpoint A

literally recalled citizens' unfamiliarity with the tasks of the water authorities in response to Statement 2. Time limitations of the interviews resulted in an unequal distribution of interview fragments over statements and viewpoints (see Figure 4). A higher coverage of interview fragments and more participants, particularly participants loading significantly on Viewpoint B and C, would likely have resulted in a more consistent image and more understanding of the participant's underlying

reasoning. Future research should consider more time for post-sort interviews or organising group discussions.

The viewpoints identified in this study are expected to be representative of water authorities in the Netherlands. Only eight out of 24 Dutch regional water authorities were included so additional viewpoints may be found if the study would be repeated with participants from the remaining water authorities. We consider this unlikely as none of the selection criteria

(see Stage 3 in the method section) were found to influence the results. Flood risk, age (years since the last organisational reform) and location were incorporated as characteristics that might influence participant's viewpoints. All viewpoints represented a mixture of age and location, thus there seems to be no relation between these characteristics and the results. Sea level might as all people with Viewpoint B were working at an authority responsible for an area below sea level. Job type did not influence the results, gender might as almost all people with Viewpoint B were male. This study cannot make

any claims regarding the influence of flood risk or gender, but this is a promising direction for further research.

A quantitative follow-up study can be used to determine the distribution of viewpoints across water authorities or within one organisation. Such a study would further justify generalisation of the results to the Dutch water authorities and would allow to check how wide-spread the overall positive attitude towards citizen science found in this research actually is. This article

presents results for practitioners in the Netherlands, but we encourage others to repeat the study, particularly in other countries facing low citizen water awareness, as described by the OECD (OECD, 2011). The developed set of statements can serve as a basis for such a study as they are not unique to the Dutch situation.

**5 Conclusion**

This study contributes to understanding about government practitioners' acceptance and perception of citizen science. A Q

methodological approach was applied to identify the viewpoints of practitioners on citizen science, in the case of water quality monitoring at Dutch regional water authorities. Water authority employees sorted a set of 46 statements related to citizen science. Three factors were identified in a factor analysis and translated into corresponding viewpoint narratives.

The first viewpoint, Viewpoint A, is named 'Citizen participation for data application'. People with Viewpoint A see more

opportunities than challenges when it comes to citizen science. They see applications in practical use of the data, but also for the active engagement of people. The second viewpoint, Viewpoint B, is named 'Water authority in control'. People with this viewpoint see a potential for data contributions by citizens in an illustrative way, but are concerned with challenges in

organisational capacity, expectation management and motivating citizens as well. The third viewpoint, Viewpoint C, is named 'Education and local knowledge'. People with this viewpoint focus on educational goals, such as teaching people about their environment and getting schools involved. They consider data applicability of secondary importance, although the data can be used illustratively.

The outcomes of this study provide strong indications that practitioners at Dutch water authorities welcome citizen science. These practitioners further believe citizen science can contribute to bridging the awareness gap as identified by the OECD (2014) and the Dutch Water Authorities (UvW, 2015a). All three viewpoints are positive towards citizen science in the form of contributory projects in which citizens collect data. People with Viewpoint A support collaborative citizen science as well, but none of the viewpoints support co-created projects between citizens and water authorities. Interviews identified low expectations in citizens' motivations and capacities as underlying causes for this low support for higher levels of citizen involvement. This may jeopardize the much-needed trust in the relation between citizens and practitioners. Although participants recognised the potential of citizen science to change governance structures, the design of citizen science projects in the Netherlands is not expected to move beyond contributory projects in the near future.

**Acknowledgements**

We thank S. Abu Shoaib and an anonymous referee for their valuable comments on the discussion paper.

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
