# Peer review of "Practitioners' viewpoints on citizen science in water management: a case study in Dutch regional water resource management"

_Hydrology and Earth System Sciences, 2016_

## Referee Comment (RC1) · S. Abu Shoaib (Referee) · 19 Oct 2016

The authors have submitted a thought-provoking paper on "Practitioners' viewpoints on citizen science in water management. Use of Q-methodological approach made the paper more interesting. I enjoyed reading the manuscript; it is very well written and easy to follow. The subject is timely and fits the journal scope well. However I believe it can be improved significantly and the message can be better presented. I recommend minor revision. Below are my comments that may be useful to the authors in updating the manuscript.

[Figure]

1. The paper discusses a very interesting aspect of citizen science in water resource management which is usually ignored. One of the most important parts of citizen science could be use of modelling simulation and prediction. In page 2, paragraph 3, line 21 of the manuscript, mentioned the process of decision making of water management. But the manuscript did not address the process development of water management that is vital for decision making. I would suggest, incorporate a complete section or a paragraph focusing recent development of modelling and prediction tools of water management. A unified approach for process-based hydrologic modeling [Clark et al., 2015] and a metric for attributing variability in modelled streamflows [Shoaib et al., 2016] could be useful reference to add in this context.

2. Applied Q methodology is subjective in nature, and findings contain possible bias. A framework to reduce bias from this approach can be outlined to strengthen the content of the paper.

3. I couldn't evaluate the result section because the figures and tables are not presented properly. Captions fail to explain the figures and tables properly. I would suggest to present the figure and tables in better ways considering P-set, Q-set and Run PQ method.

4. Table1 can be summarized through graphical interface to observe the overall agreement or disagreement in different category. Box plot or other similar plot will attribute final factor loadings better after rotation presented in Table 2.

5. The novelty of the proposed approach is not clear. Please explain exactly how the overall approach is better and effective compare to other existing methods.

6. The abstract remains quite vague about the results; more specific and quantitative results should be included in the abstract.

7. Although there is no support for higher levels of citizen engagement, how effective will be citizen science as a form of public participation? Please explain and discuss.

8. In page 2, paragraph 3, line 24 of the manuscript, mentioned "although they also indicated they doubts". Not clear enough. Please check.

9. In page 20, paragraph 2, line 5 of the manuscript"There appears to be a mismatch between the intentions of the participants and the way they trust citizens with the level of participation" – need more clear views.

10. In page 21, line 9-10,"The correlation between Factor A and C was 0.43, which indeed indicates that they are interrelated and overlap. Typically correlations above 0.39 are considered significant". What will be conclusion if the correlation between Factor A and C was 0.4 or 0.35. Please explain.

11. Understanding uncertainty as well as quantification is important in water management practice [Shoaib et al., 2016]. If the impact of uncertainty is properly explained as a form of public participation, understanding about government practitioners' acceptance and perception of citizen science will be more effective. Adding uncertainty dimension, surely add light in the proposed approach.

REFERENCES

Clark, M. P., et al. (2015), A unified approach for process-based hydrologic modeling: 1. Modeling concept, Water Resources Research, 51(4), 2498-2514.

Shoaib, S. A., L. Marshall, and A. Sharma (2016), A metric for attributing variability in modelled streamflows, Journal of Hydrology, 541, Part B, 1475-1487.

---

## Referee Comment (RC2) · Anonymous Referee #2 · 4 Nov 2016

This manuscript provides interesting insight into opinions of Dutch water management professionals about the use of citizen science as a tool that water management authorities might support. The method provided to assess collective opinions seems realistic to transfer across locations. In this case, three generalized viewpoints about citizen science that exist across 8 of the 24 water management authorities were identified. However, based upon the extreme difference in explained variance of Viewpoint A (53%) and those of Viewpoints B and C (8% and 6%), one might alternatively conclude that there is primarily only one viewpoint of participants in this study, not three. Additional explanation to support inclusion of three viewpoints would be valuable, as the difference between 6% and 1% is much less than 53% and 8%, while, instead, the

difference between 6% and 1% is currently said to demonstrate "a clear cut off after the third factor."

More specific comments follow. A number of these comments relate to words or phrases used that need further explanation, seem inappropriate, or are misleading:

*P. 2 line 18 – "lingering?" What is meant by this? *P. 2 line 31 – In reference to "(on-line) citizen science," do the authors mean studies have been carried out on projects through which people participate online? Clarify. *P. 3 second paragraph: Further clarification is needed about how "knowledge generation" differs from "raising awareness" and "public education." All are knowledge generation, are they not? *P. 3 lines 13-14 – A reference(s) is needed to support that citizen science may be used to add or test new monitoring methods. *P. 3 line 16 – What type of literature often mentions public education as an important purpose of citizen science? References needed. *P. 3 line 21 – What is meant by "early stages" of citizen science? *P 4 lines 9 and 10 – "and policy development" is included twice. Should it be included as its own purpose or is it meant to be included in connection with the other categories? *P. 4 line 14 – "This view is too limited." A reference(s) is needed. (Who says the view is too limited?) *P. 4 line 19 – Is research limited to scientists about motivations? Suggest rewording. *P. 4 line 27 – Continued attention by whom is needed? *P. 5 line 1 – What aspects of water authorities' work would benefit from citizen scientist participation? *P. 5 line 25 and P. 6 line 5 – It is unlikely that the authors could collect "all possible" opinions on the topic of citizen science. Consider softening the language about this aspect of the research. *P. 6 line 11 – In regard to the phrase "too broad," does this mean beyond the scope of the study? Further explanation or rewording recommended. *P. 6 line 12 – What were the other students' majors? Were they natural resources-related, or STEM fields, or something not at all related to water resources? Provide general information to help the reader understand the level of knowledge they might have about the statements being reviewed. *P. 16, line 6 – What is considered middle aged? Consider adding a table that describes key (i.e., those that were assessed in analyses) demographics/characteristics

of participants whose viewpoints were assessed. \*P. 18 Section 4.1 – This seems to be a continued presentation of results, not discussion. Recommend separating out results from this section and deepening the discussion. As a result, the authors will likely need to make the results section more brief; consider including phrases currently included in the results in a table or tables to shorten the text of this section of the manuscript. \*Table 3 – Recommend including a more detailed table title. For instance, is this table meant to show the level of support for each of the three viewpoints or something else? Also, consider moving "Theme 1:" and "Theme 2:" from the table description to within the table to the left of "Perception of citizen. . ." and "Acceptance of citizen. . ." respectively, for clarification. \*Table 4 – What are "applicable purposes?" Further explanation in table description would be useful. \*P. 19, line 15 – As related to "the legal obligations a water authority has regarding water quality monitoring," in regards to what? Results to action? \*P. 20 line 4 – "intentions of the participants," do the authors mean this as related to the role(s) a citizen could or should play in citizen science? \*P. 20 line 4 – Is the phrase "the way they trust" meant to indicate the level of trust of citizens by the water authorities? Rephrase for better clarity. \*P. 20 line 14 – "collide" seems an inappropriate word here. Do the authors mean this "disagrees" with Viewpoint C? \*P. 20 lines 18-19 – Was researcher bias reduced by collecting statements from various sources or by some other means? Further explanation of how researcher bias was reduced would be useful to readers. \*P. 21 line 1 – "enhancing the scope" seems an inappropriate phrase. Do the authors mean the viewpoints were broadened? \*P. 21 lines 3 and 4 – Awkward. Suggest rewording. \*P. 21 line 17 – Unsure what is meant by "on measures." What measures? \*P. 21 line 22 – Time limitations of interviews or some other type of time limitation? \*Discussion/Conclusion – Consider including discussion of the larger representation of Viewpoint A among participants as compared to Viewpoints B and C, (that is, as Viewpoints B and C explained much less of the variance in the model as compared to Viewpoint A). \*P. 22 line 29 – "Transformation of governance structures" seems broader than the subject matter of the manuscript. \*P. 22 lines 29 and 30 – The authors should mention that this statement applies to citizen science projects in NL,

not more broadly. *In general, the manuscript would benefit from a careful review of sentences, words, punctuation and grammar. Errors were made in a number of locations – this includes, but is not limited to the following: p. 4 line 11 by "wide" do the authors mean they are using the term in a broad sense?; p. 4 line 20 "compatible" should be "comparable;" check for and modify run-on sentences (e.g., p. 6 line 14); p. 6 line 22 commas are needed; data should be plural throughout the manuscript; check plural possessive apostrophes throughout (e.g., citizens' not citizen's); p. 18 line 24 – Commas should be added to identify to readers if the image of the water authority and organizational capacity go together, or if organizational capacity and lack of internal support go together; p. 21 lines 3 and 4 should be reworded; remove "especially" and "particularly" from start of sentences; the use of "we" seems appropriate for the manuscript. Use consistently throughout (rather than using third person "the authors" in some paragraphs); p. 21 lines 26-28 – Simplify this sentence.
* * *

---

## Author Comment (AC1) · 6 Nov 2016

We thank the reviewer for her/his constructive comments. Below we list our thoughts on improving the manuscript. We appreciate any additional feedback.

1. We think it is indeed appropriate to add a paragraph on the use of modelling and prediction tools in water management. We will do so in the revision of the article.

2. We have discussed how to make the steps taken to reduce bias more explicit and present them as a framework in the revised article. We plan to present this as follows:

The bias introduced is twofold:

[Figure]

a. Researcher bias: the opinion of the researcher is too dominant

b. Selection bias: the selection of the statements (Qset) or the participants (Pset) is biased and over representing one perspective.

Researcher bias can be introduced in the formulation of the statements, selection of the participants and the interpretation of the results. Following actions were taken to reduce bias in each of these steps:

a1 Formulation statements (Qset): following Watts & Stenner (2000) we based the statements where possible on quotes from others.

a2 Selection participants (Pset): we did not know the participants except for two, so we also did not know their opinion beforehand.

a3 Interpretation: we followed a structured way of describing the viewpoints based on distinguishing statements (Watts & Stenner, 2000) to avoid cherry picking.

Selection bias can appear in the selection of the Qset and Pset. Following actions were taken to reduce bias:

b1 Formulation Qset:

b11 Base the Qset on a diverse set of interviews to avoid dominance of one perspective.

b12 Ask each participant (Pset) if they miss a statement to check if all relevant aspects were covered and nothing new came up.

b2 Selection Pset:

b21 Apply snowball sampling. Ask participants to suggest potential participants with other opinions.

b22 Have participants with a diverse background and job description. If desirable we can add a table with background and job description of the participants.

In the discussion, we may add the note that further bias reduction can be achieved by having two or more researchers execute each step in parallel.

3. We will improve the captions in the revision.

4. Thanks for this suggestion. We can colour-code the loadings to make is easier for readers to quickly see agreement and disagreement in Table 1. We think we can improve the readability of Table 2 by replacing this table with a radar chart.

5. We will improve the explanation of the novelty of the approach in the introduction and discussion. We see three main novelties :

a. Scarcity of research on the motivations of the target group, compared to citizens and scientitst.

b. Linking the (envisioned) use of citizen science to (envisioned) level of citizen participation

c. With respect to the methodology: linking the qualitative and quantitative aspects by using Q methodology. So far this has not been applied to this problem. Our approach has the advantage over quantitative approaches in that we have explored a wide spectrum of viewpoints whereas quantitative methods would arrive easily ad averaged values. Quantitative methods will likely not reveal the distinguishing elements that are in our opinion crucial to be aware of for effective implementation of citizen science. Our approach has the advantage over fully qualitative methods that it reduces the variation in opinions to a representative small set of viewpoints in a trackable, relatively little biased way (see above). The fact that the number of viewpoint is small makes them useful for actual implementation of citizen science campaigns.

6. We will make the abstract more specific in the revision.

7. We have read about two studies that describe citizen science as a form of public participation. One is about air pollution (http://sth.sagepub.com/content/35/2/244.short). The other case related to water quality in Rhode Island is mentioned in the review

of Conrad & Hilchey. We will add a reference to these cases in the introduction and discussion in the revision.

---

## Author Comment (AC2) · 13 Nov 2016

We thank the reviewer for his/her critical assessment of our work and his/her useful suggestions.

The most important discussion point is the number of factors/viewpoints that we selected. We will try to clarify this aspect in the revised manuscript. Points we plan to include in this explanation are:

The objective of our study was to explore potentially different viewpoints and that motivated the use of Qmethodology. We sought to explore the variation in viewpoints in our population rather than find the opinion of the majority, which is typically done with

quantitative questionnaires.

In our understanding, the explained variance in factor analysis should be considered cumulative. With Factor A we explain 53% of the variance in the sample, with Factor A and B 71%, and with Factor A-C 77%. Inclusion of more factors gives only minor rise in explained variance. Put in another way if we would have 100 marbles, 53 would fall in box 1, 8 in box 2, 6 in box 3 and no groups can be found among the other marbles.

Essential motivation for three factors followed from combination of quantitative and qualitative results. We found that Factor B and C are not part of A. The high correlations among factors are due to the high number of consensus statements. Yet our qualitative analysis supported the inclusion of 3 viewpoints. The differences between A and C particularly regarding if participants think Citizen Science is merely useful for Education (Viewpoint C) or a combination of education & data collection (Viewpoint A) is very relevant, as for citizens the actual use of their data is key for success (also mentioned in the Discussion).

We responded to the detailed comments below: -P. 2 line 18 – "lingering?" What is meant by this?

—"on the verge of breakthrough", we will clarify this in the revision

-P. 2 line 31 – In reference to "(on- line) citizen science," do the authors mean studies have been carried out on projects through which people participate online? Clarify.

—"citizen science in online and in field projects", we will clarify this in the revision

-P. 3 second paragraph: Further clar- ification is needed about how "knowledge generation" differs from "raising awareness" and "public education." All are knowledge generation, are they not?

—The distinction we try to make here is between "just" educating the public and generating actual new knowledge about the system or problem at hand by participation of the public. We will define these terms more clearly in the revision.

-P. 3 lines 13-14 – A reference(s) is needed to support that citizen science may be used to add or test new monitoring methods.

—We base this statement on the explorative interviews held at the start of this work. We will include this reference in the revision.

-P. 3 line 16 – What type of literature often mentions public edu- cation as an important purpose of citizen science? References needed.

— We will add references for this statement in the revision (e.g. Cohn (2008))

-P. 3 line 21 – What is meant by "early stages" of citizen science?

—"early stages" refers to policy development in the previous sentence → "In early stages of the policy development process, citizen science can be used to discover alternative management actions." We will clarify this in the revision.

-P 4 lines 9 and 10 – "and policy development" is included twice. Should it be included as its own purpose or is it meant to be included in connection with the other categories?

—this was a mistake we will change this in the revision.

-P. 4 line 14 – "This view is too limited." A reference(s) is needed. (Who says the view is too limited?)

—It is our opinion. We will motivate this better and change the sentence to "We find this view too limited, . . ."

-P. 4 line 19 – Is research limited to scientists about motivations? Suggest rewording.

—We will rephrase this in the revision "research on professional's motivations"

-P. 4 line 27 – Continued attention by whom is needed?

—We will specify this in the revision as "Continued attention by those managing citizen science projects . . ."

-P. 5 line 1 – What aspects of water authorities' work would benefit from citizen scientist participation?

—The cited articles are mainly considering flood risk management. We will specify that in the revision.

-P. 5 line 25 and P. 6 line 5 – It is unlikely that the authors could collect "all possible" opinions on the topic of citizen science. Consider softening the language about this aspect of the research.

— We will soften this in the revision as P5/25: "The first stage aimed to collect as many opinions as possible"; P6/5: "... (all identified statements on a topic) ..."

-P. 6 line 11 – In regard to the phrase "too broad," does this mean beyond the scope of the study? Further explanation or rewording recommended.

—We will reword this in the revision: Too broad = open to multiple interpretations or "ambiguous"; "off-topic" means indeed "out of scope"

-P. 6 line 12 – What were the other students' majors? Were they natural resources-related, or STEM fields, or something not at all related to water resources? Provide general information to help the reader understand the level of knowledge they might have about the statements being reviewed.

— We will add this information in the revision. The majors of the other students were medicine, mathematics, mechanical engineering and management studies. We avoided jargon in the statements and indeed most statements were clear to all students. Only the term "water awareness" raised a question among the students, yet this term is common language among the Pset.

-P. 16, line 6 – What is considered middle aged? Consider adding a table that describes key (i.e., those that were assessed in analyses) demographics/characteristics of participants whose viewpoints were assessed.

—Middle-aged was defined as between the age of 45 and 65. We can add a reference to same age categories used in other citizen science research. We are hesitant to put emphasis on participant characteristics. This may give a reader the impression that we checked if these characteristics had any influence. We only mention the characteristics to warrant the quality of our research, to demonstrate that our sample (Pset) was diverse enough to get a wide range of opinions. Research into the link between characteristics Investigating the link between people characteristics and options was outside of the scope of this research and could better done with a quantitative research method. We can add this more explicitly as a recommendation in the discussion.

-P. 18 Section 4.1 – This seems to be a continued presentation of results, not discussion. Recommend separating out results from this section and deepening the discussion. As a result, the authors will likely need to make the results section more brief; consider including phrases currently included in the results in a table or tables to shorten the text of this section of the manuscript.

—We will deepen the discussion with suggestions provided elsewhere in this review, such as the explained variance and seek to make a more clear separation of results and discussion. Yet we do not think that it is wise to move this whole section to the results section as that would imply presenting result interpretation using theory in the results section. We think that is more appropriate in the discussion.

-Table 3 – Recommend including a more detailed table title. For instance, is this table meant to show the level of support for each of the three viewpoints or something else? Also, consider moving "Theme 1:" and "Theme 2:" from the table description to within the table to the left of "Perception of citizen. . ." and "Acceptance of citizen. . ." respectively, for clarification.

—We will add a more "self-explicatory" table title in the revision

-Table 4 – What are "applicable purposes?" Further explanation in table description would be useful.

—We will add a more "self-explicatory" table title in the revision including the text "purposes that were considered suitable by the viewpoints" instead of applicable

-P. 19, line 15 – As related to "the legal obligations a water authority has regarding water quality monitoring," in regards to what? Results to action?

—We will clarify this in the revision as "…regarding the use of standardized methods and reports on water quality monitoring"

-P. 20 line 4 – "intentions of the participants," do the authors mean this as re- lated to the role(s) a citizen could or should play in citizen science?

—We will clarify this in the revision "intentions of the participants regarding the purposes citizen science should support…"

-P. 20 line 4 – Is the phrase "the way they trust" meant to indicate the level of trust of citizens by the water authorities? Rephrase for better clarity.

—-We will rephrase this in the revision as " …and the expressed level of trust in citizens as reflected by the level of envisioned citizen participation."

-P. 20 line 14 – "collide" seems an inappropriate word here. Do the authors mean this "disagrees" with Viewpoint C?

— We will use a more neutral wording in the revision such as "Another important motivation for citizens is the provision of feedback on how the data is used (e.g. Bonney et al., 2009; Rotman et al., 2012; Roy et al., 2012), which is particularly in contrast with opinions in Viewpoint C."

-P. 20 lines 18-19 – Was researcher bias reduced by collecting statements from various sources or by some other means? Further explanation of how researcher bias was reduced would be useful to readers.

—This was the main concern of reviewer 1. We proposed a framework for reducing researcher bias in the response to reviewer 1 that we plan to include in the revision.

-P. 21 line 1 – "enhancing the scope" seems an inappropriate phrase. Do the authors mean the viewpoints were broadened?

— Yes we will rephrase this in the revision.

-P. 21 lines 3 and 4 – Awkward. Suggest rewording.

—We will reword this in the revision: "At several occasions the research was presented to water authority employees. They were surprised that all viewpoints were relatively positive about citizen science, because they believed to have colleagues who are indeed more skeptical than these three viewpoints."

-P. 21 line 17 – Unsure what is meant by "on measures." What measures?

—We will specify this in the revision as "measures taken to realize policy goals."

-P. 21 line 22 – Time limitations of interviews or some other type of time limitation?

—Time limitations of the interviews indeed. We will specify that in the revision.

-Discussion/Conclusion – Consider including discussion of the larger representation of Viewpoint A among participants as compared to Viewpoints B and C, (that is, as Viewpoints B and C explained much less of the variance in the model as compared to Viewpoint A).

—We will address this in the discussion. We will explain that no generalizations can be drawn from that Viewpoint A explains most of the variance. It should not be interpreted as that most water professionals agree with viewpoint A.

-P. 22 line 29 – "Transformation of governance structures" seems broader than the subject matter of the manuscript.

—We will rephrase this claim in the revision

-P. 22 lines 29 and 30 – The authors should mention that this statement applies to citizen science projects in NL, not more broadly.

—We will specify this in the revision.

-In general, the manuscript would benefit from a careful review of sentences, words, punctuation and grammar. Errors were made in a number of loca- tions – this includes, but is not limited to the following:

p. 4 line 11 by "wide" do the authors mean they are using the term in a broad sense?;

p. 4 line 20 "compatible" should be "comparable;" check for and modify run-on sentences (e.g., p. 6 line 14);

p. 6 line 22 commas are needed; data should be plural throughout the manuscript; check plural possessive apostrophes throughout (e.g., citizens' not citizen's);

p. 18 line 24 – Commas should be added to identify to readers if the image of the water authority and organizational capacity go together, or if organizational capacity and lack of inter- nal support go together;

p. 21 lines 3 and 4 should be reworded; remove "especially" and "particularly" from start of sentences; the use of "we" seems appropriate for the manuscript. Use consistently throughout (rather than using third person "the authors" in some paragraphs); p. 21 lines 26-28 – Simplify this sentence.

—We will review the manuscript carefully and improve the writing.